# Algorithmic Assessments in Deciding on Voluntary, Assisted or Involuntary Psychiatric Treatment

**DOI:** 10.3390/diagnostics12081806

**Published:** 2022-07-26

**Authors:** Gerhard Grobler, Werdie Van Staden

**Affiliations:** 1Department of Psychiatry, Faculty of Health Sciences, University of Pretoria, Pretoria 0002, South Africa; gerhard.grobler@up.ac.za; 2Centre for Ethics and Philosophy of Health Sciences, Faculty of Health Sciences, University of Pretoria, Pretoria 0002, South Africa

**Keywords:** mental capacity, informed consent, mental incompetence, algorithms, medical legislation, decision support techniques

## Abstract

The challenges in assessing whether psychiatric treatment should be provided on voluntary, assisted or involuntary legal bases prompted the development of an assessment algorithm that may aid clinicians. It comprises a part that assesses the incapacity to provide informed consent to treatment, care or rehabilitation. It also captures the patient’s willingness to receive these treatments, the risk posed to the patient’s health or safety, financial interests or reputation and risks of serious harm to self or others. By following various decision paths, the algorithm yields one of four legal states: a voluntary, assisted, or involuntary state or that the proposed intervention should be declined. This study examined the predictive validity and the reliability of this algorithm. It was applied 4052 times to 135 clinical case narratives by 294 research participants. The legal states yielded by the algorithm had high statistical significance when matched with the gold standard (Chi-squared = 6963; df = 12; *p* < 0.001). It was accurate in yielding the correct legal state for the voluntary, assisted, involuntary and decline categories in 94%, 92%, 88% and 86% of the clinical case narratives, respectively. For internal reliability, a correspondence model accounted for 99.8% of the variance by which the decision paths clustered together fittingly with each of the legal states. Inter-rater reliability testing showed a moderate degree of agreement among participants on the suitable legal state (Krippendorff’s alpha = 0.66). These results suggest the algorithm is valid and reliable, which warrant a subsequent randomised controlled study to investigate whether it is more effective in clinical practice than standard assessments.

## 1. Introduction

Clinicians are ordinarily required to make a clinical assessment of patients with mental illness in terms of mental health legislation, especially for purposes of involuntary treatment and hospitalisation. This is required by The Principles for the Protection of Persons with Mental Illness and the Improvement of Mental Health Care, as adopted by the United Nations General Assembly in 1991 and the mental health legislation of most countries [1,2]. This clinical assessment requires not only an evaluation of the clinical features and the making of a provisional diagnosis but also requires the application of the stipulations of mental health legislations.

Both the clinical aspects and legal prescriptions of this assessment are rather difficult [3,4]. There are many reasons for this difficulty, including the complexity in the clinical presentation of the patient, difficulty in interacting with the patient (for example, in case of psychosis), translating the stipulations captured in the language of law and regulations into practice, capacity assessments, conflicting values and ethical obligations, and an assessment context in which there is a high risk of serious or life-threatening harm to the patient or other people [3,4,5,6,7,8,9]. Adding to these difficulties, front-line clinicians who must make these assessments at the coalface are often not initially psychiatrists but clinicians at emergency departments, family physicians and general practitioners [3,4,9].

To aid clinicians in this rather difficult clinical assessment, an algorithm may improve accuracy in decisions and reduce the risk of harm [10]. Whilst using diagnostic algorithms and prediction models is well-established in medicine [11,12], no validated algorithm exists as far as we could establish for assessing whether psychiatric treatment should be provided on voluntary, assisted or involuntary legal bases. In response, we developed an algorithm that captures the legal requirements of the Mental Health Care Act (MHCA) no. 17 of 2003 in South Africa [13]. It operationalises the requirements of the MHCA by which all treatments, care or rehabilitation (TCR) of mental health care users (MHCU) should be rendered in accordance with an assessment of an individual’s legal state as being a voluntary, involuntary or an assisted MHCU. Hospitalisation is included within the scope of TCR but the three legal states also apply outside a hospital setting. The required assessment is in pursuance of the main aim of the MHCA by which mental health services must be provided in the least restrictive ways possible. In addition to its legal imperatives, the assessment is ethically crucial for preserving the rights of individuals by which they may make decisions on their mental healthcare autonomously when they have the capacity to do so; on the other hand, it also ensures that individuals who are incapable of giving consent to mental healthcare may still be provided with treatment, care or rehabilitation. Pivotal in this assessment is, thus, whether an individual is incapable of giving consent to the proposed treatment, care or rehabilitation owing to a mental illness.

The Incapacity and Legal State Assessment Algorithm (ILSAA) comprises accordingly a part that assesses incapacity to give consent to the proposed TCR. It also captures the patient’s willingness to receive TCR, the risk posed to his or her health or safety, the risks of serious harm to self or others or the financial interests or reputation of the patient. Through various decision paths, the ILSAA yields one of four possible legal states as being suitable: voluntary, assisted, involuntary or that TCR is declined. The ILSAA is available as Appendix A from which the requirements of the MHCA may be scrutinised in addition to the MHCA itself. The ILSAA derived its content validity from the stipulations of the MHCA and the literature on incapacity assessment [7,13,14,15], but this kind of validity would not be sufficient in answering the research questions on whether it would practically assess that which it is supposed to assess, and whether it would do so consistently. Addressing these questions, the study aimed to test the predictive validity and the reliability of the algorithm.

## 2. Methods

### 2.1. Design

The ILSAA was applied 4052 times to 135 clinical case narratives by 317 research participants. Its predictive validity was tested by comparing the legal states as yielded by the ILSAA assessments, with the gold standard for each clinical case narrative. This included the potential influence of clinical features on the accuracy of the ILSAA. For internal reliability testing, the four legal states yielded by the ILSAA were tested for an association with the ILSAA’s decision paths. Moreover, the ILSAA was examined for internal consistency among its decision paths. For inter-rater reliability, consistency between participants was examined for the legal states yielded by the ILSAA and its decision paths.

### 2.2. Research Participants

From the population who are meant to use the algorithm, two groups were conveniently sampled: psychiatrist-in-training who were already registered health care practitioners as designated by the MHCA and general physicians-in-training who would be qualified as health care practitioners in the subsequent year or two. Unlike some countries, all psychiatrists-in-training, called registrars in psychiatry, first had to complete training as general physicians and be professionally registered with the Health Professions Council of South Africa as medical practitioners. They were all enrolled in the postgraduate degree programme in psychiatry at the University of Pretoria, South Africa, and appointed for executing clinical duties in a specialist clinical internship post at a large psychiatric hospital. The general physicians-in-training were enrolled at the same university and were all within 10 to 20 months of completing a six-year medical degree (i.e., MBChB).

### 2.3. Clinical Case Narratives

The set of clinical case narratives on which the ILSAA was applied captures a variety of potential clinical presentations, various symptom permutations, subjective intensity in emotional experience and suicide risk indicators [16]. The set narrates clinical cases to which all four legal states apply. These are voluntary, assisted or involuntary TCR, as well cases in which TCR was not clinically indicated or it was not a priority even when clinically indicated. In addition, some narratives are relating cases for which sufficient diagnostic criteria were not met to diagnose a mental disorder.

The set of clinical case narrative presents various symptom permutations that include features of mood disorders, cognitive symptoms, psychotic symptoms, the extent and duration of symptoms, the intensity and qualities of afflicted emotions and what the experiences were about. Some narratives relate cases of incongruence between the patient’s account of severity and that of others (e.g., the physician or family) and different points of view on the suicide risk.

The clinical case narratives contain features related to suicidality (54% of cases), cognitive impairment (35% of cases), psychotic symptoms (27% of cases), incongruence between the severity of symptoms as evaluated by the patient and others (18% of cases) and incongruence in the evaluation of suicide risk (24% of cases). The features related to suicidality include a family history of completed suicide, past or recent suicide attempts, past or current suicidal thoughts, suicide planning and current non-suicidal self-injurious behaviour.

Each clinical case narrative provides information that is required for assessing incapacity owing to mental illness, whether the person is willing or unwilling to receive the proposed CTR, the health or safety of the person and/or that of other people, potential or actual serious harm to himself or herself and/or other people and, when pertinent to the circumstances, the person’s financial interests, dignity or reputation.

The clinical case narratives were purposively developed for the training of physicians-in-training and mental health practitioners in applying the MHCA. An example is available in Appendix A. Using the Delphi-method, a panel of three experts who were all senior academics and psychiatrists evaluated each clinical case narrative for being authentic and clinically credible. They also independently assigned the most suitable of the four legal states to each of the clinical case narratives. They had first made the assignments individually and then sought consensus through a process of discussion on each narrative. To achieve 100% agreement among the panel members, ambiguous narratives were discarded and narratives were refined to ensure that the gold standard legal state was as unequivocal as was possible for each case narrative. The panel did not use and had no prior exposure to the algorithm in assessing the gold standard outcomes.

### 2.4. Variables

The variables examined in the study were the four legal states yielded by the ILSAA and the gold standard legal state for each clinical case narrative. The legal states were whether the MHCU should receive voluntary, assisted or involuntary TCR. The fourth category was whether mental health TCR should be declined when it was not clinically indicated or not a mental health service priority.

Further variables were the 56 decision paths of the ILSAA that could be followed through 36 potential decision points, resulting in one of the four legal states. There were 12 decision paths for the voluntary, 9 for the assisted, 20 for the involuntary and 15 for the declined legal state categories.

The clinical features that were examined as variables were psychotic symptoms, cognitive impairment, suicidality, incongruence between the accounts of the patient and that of others regarding the severity of the patient’s condition and incongruence between the accounts of the patient and others on the suicide risk.

### 2.5. Procedures

Three hundred general physicians-in-training and 17 of a potential 20 psychiatrists-in-training (85%) participated in training sessions during February and March 2020 on the MHCA’s requirements and the use of the algorithm. The sessions entailed an introductory lecture on the MHCA and demonstration of the algorithm by applying it interactively to four example case narratives (that were not used in the study). Thereafter, each trainee was requested to apply the algorithm to a set of 15 randomly selected clinical case narratives that was randomly assigned to each trainee. The trainees marked the suitable decision path and legal state for each of the clinical cases assigned to them. Some trainees completed fewer than 15 cases owing to the time constraints of these training sessions.

The trainees who participated in the training sessions were invited to participate in this ethically approved study. Only the data of trainees who provided written informed consent to participation in the study were then collated for analyses. Twenty-three of the general physicians-in-training (7.7%) in the training sessions and three of 20 psychiatrists-in-training (15%) declined the invitation to participate in the study.

### 2.6. Statistical Analyses

The predictive validity of the algorithm was examined by calculating the goodness-of-fit between the legal states yielded by the ILSAA and the gold standard legal states, using the chi-squared test and standardised residuals. Moreover, the predictive accuracy, sensitivity, specificity, precision rate or positive predictive value (PPV), the negative predictive value (NPV), the miss rate or false negative (FN) rate and the fall-out or false positive (FP) rate were calculated. Since multiple participants (and, thus, ILSAA applications) rather than a singular test contributed to each case narrative’s evaluation as true or false positive and true or false negative, all the ILSAA applications on the same case narrative were calculated as a fraction of one. The number of true positives (TP) were calculated as the sum of the fractions for all case narratives. For example, 38 of 41 applications of the algorithm on the case narrative labelled FeCo correctly identified the legal state as involuntary. This 38/41 fraction (or 0.927) was added to the fractions calculated in the same way for all the other case narratives in the involuntary legal state category, thus deriving a composite number of TPs for the involuntary legal state category. The same was performed for all the legal states in calculating true negative (TN), FP and FN values. The composite fractions were rounded to obtain integers as is standard practice for expressing TP, TN, FP and FN values.

Whilst predictive accuracy concerns all instances of TPs and TNs using the legal states as points of reference, the case-specific accuracy calculations took each case narrative as a point of reference. The case-specific accuracy is the proportion of ILSAA applications that was accurate for a specific case narrative. For example, when 30 of 40 applications of the ILSAA were the same as the gold standard for a particular case, the case-specific accuracy was 75%. The mean and median for case-specific accuracies were calculated. The mean of the case-specific accuracies is an alternative method of expressing the ratio between TPs and the total number of case narratives. Pearson’s correlations were calculated between the case-specific accuracies and the clinical variables.

For internal reliability testing, Pearson’s chi-squared test was used to examine the association between the legal states and the decision paths. In addition, a correspondence analysis was performed, which was modelled using the symmetric normalisation of the legal states and the decision paths along two dimensions.

For inter-rater reliability, the extent of agreement between the participants in applying the ILSAA was calculated using Krippendorff’s alpha coefficient, which provides for a categorical metric across variable numbers of applications of the ILSAA on each clinical case narrative and that each participant could apply the ILSAA only to some of the case narratives. Potential Krippendorff’s alpha values range from −1 to 1, that is, from perfect systematic disagreement to perfect systematic agreement, with 0 indicating the absence of both agreement and systematic disagreement. Bootstrapped sampling with 95% confidence intervals was performed on the Krippendorff’s alpha calculations, with the advantage of adjusting for data not following a normal distribution.

The probability threshold for a type I error was set at 5%. SPSS version 27 was used for the analyses.

## 3. Results

### 3.1. Descriptive Features

The 294 participants comprised 277 general physicians-in-training and 17 psychiatrists-in-training. These groups, respectively, represent 92.2% and 85% of the trainees who had been requested to participate in the study. The participants applied the ILSAA 4,052 times. Six of these applications had no legal state outcome recorded, and for fifty-five (1.4%), the decision path was marked ambiguously. For example, a participant made corrections without clearly indicating some of the decision points.

Table 1 presents the number of case narratives for each of gold standard legal state outcomes. Owing to the random distribution of clinical case narrative among participants, the number of times that the ILSAA was applied to each clinical case varied. The ILSAA was applied 9 to 41 times for each of 131 case narratives. Four of the initial one-hundred and thirty-five case narratives were excluded from analysis, since fewer than seven participants applied the ILSAA to these cases. Table 1 also shows the average number of participants that applied the ILSAA to the case narratives in each of the legal state categories.

### 3.2. Goodness-of-Fit between the ILSAA Outcomes and Gold Standard Legal States

Supporting the predictive validity of the algorithm, the Pearson’s chi-squared test value for the association between the algorithm-derived assessments and the gold standard legal states was highly statistically significant (Chi-squared = 6 963; df = 12; *p* < 0.001), and this was the case similarly for the general physicians-in-training (Chi-squared = 6 650; df = 12; *p* < 0.001) and the psychiatrist-in-training (Chi-squared = 318; df = 9; *p* < 0.001). The covariance of the legal states yielded by the ILSAA and the gold standards for each case narrative was indicated by the large, positive standardised residuals for each of the four legal states, shown in Table 2.

The remainder of the cross-matched legal states indicate negative standardised residuals. These provide a relative indication of where the ILSAA outcomes did not match with the gold standard. The legal state most likely to be mismatched was the declined state (indicated by residuals closest to zero) when compared to each of the other legal state outcomes. The value closest to zero (that is, −3.4) is nonetheless of a magnitude that indicates independent variance [17].

### 3.3. Predictive Validity Calculations

The True Positive, False Positive, True Negative and False Negative values are shown in Table 3. Supporting the predictive validity of the ILSAA, the predictive validity calculations shown in Table 4 indicate that the ILSAA predicted the correct legal state with much accuracy (≥86%) and specificity (≥89%). The ILSAA predicted the voluntary legal state best, and it was the weakest in its sensitivity to predict the decline category. Calculations of the predictive accuracy for psychiatrists-in-training and the general physicians-in-training applying the ILSAA were within 1-percentage point of each other, except for a 2-percentage point lower value obtained for the assisted legal state among the psychiatrists-in-training.

### 3.4. Case-Specific Accuracy

The median of the case-specific accuracies was 85.7%. The mean of the case-specific accuracies for all the case narratives was 79.9%. The 95% confidence interval for the case-specific accuracies ranged from 60.8% to 99%. As shown in Table 5, correlations between the clinical features and the case-specific accuracy were weak with all coefficients being equal or lower than 0.208.

### 3.5. Internal Reliability

For internal reliability testing, the association between the legal states yielded by the ILSAA and its decision paths was highly significant statistically (Pearson’s chi-squared = 12,277; df = 366; *p* < 0.001). The correspondence analysis yielded a Cronbach’s alpha value of 0.998, which accounted for 99.801% of the variance and an Eigenvalue of 1.996. The correspondence model was highly significant statistically (Chi-square = 11,949; df = 183; *p* < 0.001). In this model, using symmetric normalisation, each legal state and the decision paths were scored on two dimensions. The scores are shown in Table 6. These scores represent a clear pattern in which the decision paths clustered together fittingly with the legal states, as shown in Figure 1.

In addition to reflecting the four legal states, Figure 1 also shows that the scores of the voluntary and decline legal states (and their respective decision paths) are each situated in an own quadrant. The scores of the involuntary and the assisted legal states (and their respective decision paths) are located in the same quadrant. Sharing the same quadrant may be reflecting that both these legal states entail that patients are incapable of giving informed consent.

### 3.6. Inter-Rater Reliability

The inter-rater reliability of the algorithm is reflected in the Krippendorff’s alpha values shown in Table 7. A Krippendorff’s alpha value between the 95% confidence interval of 0.637 and 0.694 indicates a moderate to strong agreement among the participants in applying the algorithm. The alpha value for psychiatrist-in-training was higher when bootstrapping was applied as bootstrapping corrects for the limited data (*n* = 17) by constructing the normal distribution that pertains to the data by resampling the data 1000 times.

## 4. Discussion

The results indicate that the Incapacity and Legal State Assessment Algorithm is valid in that it performed as it was supposed to. In 294 participants applying the algorithm 4052 times on a set of 135 clinical case narratives, it yielded legal states that matched excellently with the gold standard legal states. The algorithm was accurate in yielding the correct legal state for the voluntary, assisted, involuntary and decline categories in, respectively, 94%, 92%, 88% and 86% of the clinical case narratives. Its specificity ranged from 89% to 96%, and its sensitivity ranged between 82% to 89% except for a 59% sensitivity in predicting the decline category.

Results of the reliability testing indicate that the algorithm performed consistently for its decision paths and among those who applied of the algorithm. Internal consistency was found in a significant association between the legal states and the decision paths. Moreover, a correspondence analysis that accounted for 99.8% of the variance showed that the decision paths clustered clearly together for each of the legal states along two dimensions. Inter-rater reliability testing of the algorithm showed a moderate to strong degree of agreement among the participants for the legal states and modest agreement for the decision paths.

Considering the challenging, difficult and non-exact nature of the decisions captured by the algorithm [3,5,7,8], the predictive validity of the algorithm turned out to be secure by common standards and more so than we had anticipated. For the legal state category by which psychiatric services should be declined, the algorithm performed similarly than for the other legal states in terms of accuracy and specificity, but it was weaker in terms of sensitivity. In other words, the decline category was mistakenly identified as if pertaining to a clinical case narrative to approximately the same extent as the other legal states (that is, in about 5% of cases), but the decline category was mistakenly missed as most suitable (that is, in 46% of cases) more so than the other legal states that were ranging from 11% to 18% of cases. This means that the participants would have offered a psychiatric service in 46% of cases where this had not been clinically indicated or a priority. This result may reflect a contextual bias by which clinicians feel they have to offer a service, more so than reflecting an inherent quality of the algorithm.

The challenging, difficult and non-exact nature of the decisions captured by the algorithm also limited the reliability results. Whilst the internal consistency was excellent by common standards, the inter-rater reliability results reflected the difficult nature of the decisions. The modest agreement for the decision paths was mitigated by a moderate to strong degree of agreement for the legal states. This makes sense considering that several algorithm paths lead to the same legal state. This limited result should be expected in that differences in emphasis are likely, for example, regarding which aspect is most striking when incapacity to give informed consent cuts across various actions (including understanding, choosing decisively, communicating and accepting the need for an intervention) [7]. Whether specific clinical features add to this difficulty was not clear from the results in that no more than weak correlations were found between identifying the correct legal state for cases and cases featuring suicidality parameters, cognitive or psychotic symptoms.

Being the first validation study of an algorithm that assesses whether psychiatric treatment should be provided on voluntary, assisted or involuntary legal basis, results may at best be compared with the performance of dissimilar diagnostic assessment instruments. For example, the accuracy, specificity and sensitivity of the ILSAA compare favourably with the Structured Clinical Interview for DSM-5-Clinician Version (SCID-5-CV) [18], algorithms to differentiate between autistic spectrum disorder and other clinically relevant psychiatric and developmental disorders [19] and a diagnostic screening algorithm for eating disorders [20]. The validity results of the ILSAA are, furthermore, well-above the threshold of 0.70 set for health status instruments [21].

There are other limitations pertaining to the results, which may be addressed in further research. That the algorithm is valid and reliable does not mean that it is necessarily more effective than not using it at all. A subsequent study on the algorithm’s effectivity may build on the validity and reliability results, and compare it with standard assessments. To this end, a randomised controlled design will control for common factors that may influence the algorithm’s performance.

Using clinical case narratives in this study was advantageous in various respects but the use of these only partially reflects the properties of the algorithm as these would be when using the algorithm among patients. The clinical case narratives captured a large variety of permutations in which clinical features may present and it captured ways in which patients may be incapable of consenting to interventions. For validity and reliability testing, it was important that the algorithm would be tested for having sufficient reach across this variety of presentations, whereas a study among patients would need many more participants and have taken many years so as to include most of the permutations in clinical presentations. Another advantage was that the clinical case narrative presented all the information required in the assessment. This meant that the availability of information was not an issue in addition to the challenges and difficulties in applying the algorithm, as it would likely be when applying the algorithm among patients. Although the credibility and authenticity of clinical case narratives have been carefully nurtured, subsequent studies that test the algorithm among patients are nonetheless recommended.

Further limitations to the results are that the physician-in-training participants lacked in clinical experience and used the algorithm as individuals. Although the results for psychiatrists-in-training and the general physicians-in-training appeared to be similar, the small number of psychiatrist-in-training who were more experienced participants did not allow for statistical comparisons. One may hypothesise, nonetheless, that the algorithm would yield stronger results in subsequent studies when used by more experienced clinicians. Furthermore, the trainees applied the algorithm as individuals. In contrast, the MHCA requires that at least two clinicians make the assessment for the involuntary and assisted legal states. A subsequent study may examine whether the conjoint application of the algorithm would yield stronger results.

The algorithm is meant to assist and enable clinicians in making challenging, difficult and non-exact assessments of the most suitable legal state for providing psychiatric treatments including hospitalisation. However, the contents of the algorithm were derived from requirements of MHCA in South Africa [13], and it may need adapting for other jurisdictions. To this end, the algorithm may serve as an exemplar. Furthermore, its utility is restricted to being an aid. It is not a replacement for good process and meeting ethical requirements when making these clinical assessments [7].

## 5. Conclusions

An algorithm may enable clinicians in assessing patients with mental illness in terms of mental health legislation, especially for the purposes of involuntary treatment and hospitalisation. An exemplar of such is the Incapacity and Legal State Assessment Algorithm, which in this study was found to be valid and reliable in identifying whether psychiatric treatment, care or rehabilitation should be provided on a voluntary, assisted or involuntary basis or be declined by the requirements of the Mental Health Care Act in South Africa. Supported by its validity and reliability, the algorithm may be used in practice at the clinical coalface where the assessments have proven to be challenging [3,4,7]. It may also be used in clinical training. Notwithstanding its utility in practice and clinical training, the results on the validity and reliability of the algorithm further warrant subsequent randomised controlled studies on whether it is more effective in practice and training than standard assessments and training methods.

## Figures and Tables

**Figure 1 diagnostics-12-01806-f001:**
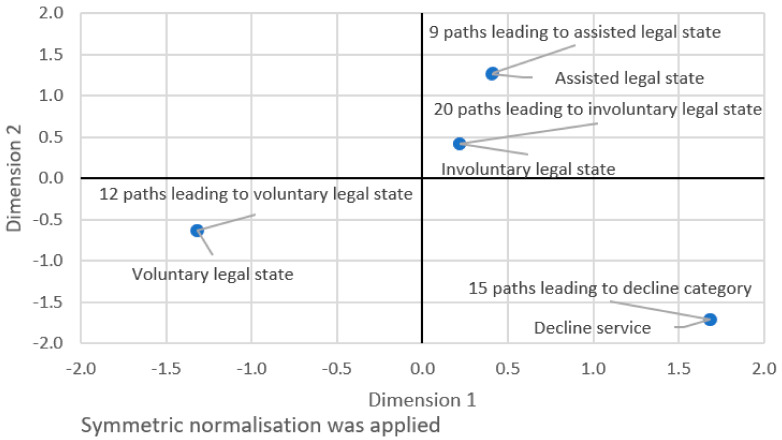
Correspondence model between legal states and decision paths.

**Table 1 diagnostics-12-01806-t001:** Gold standard legal states of the case narratives and mean number of participants applying the ILSAA per case narrative.

Gold Standard Legal State	Number of Case Narratives	Mean Number of Participants Applying the ILSAA Per Case Narrative
Voluntary	37	35.5 (standard deviation = 6.6)
Involuntary	33	31.9 (standard deviation = 8.8)
Assisted	32	33.8 (standard deviation = 8.6
Decline	29	21.0 (standard deviation = 10.2)

**Table 2 diagnostics-12-01806-t002:** Standardised residuals comparing the legal states yielded by the ILSAA outcomes to the gold standard legal states.

Legal State			ILSAA Outcomes				Total
			Assisted	Declined	Involuntary	Voluntary	
Gold Criterion	Assisted	Count	894	23	124	38	1081
		Expected count	272.4	154.7	318.8	334	1081
		Standardised residual	37.7	−10.6	−10.9	−16.2	
	Declined	Count	36	383	134	56	609
		Expected count	153.5	87.2	179.6	188.2	609
		Standardised residual	−9.5	31.7	−3.4	−9.6	
	Involuntary	Count	35	80	933	2	1051
		Expected count	264.8	150.4	310	324.7	1051
		Standardised residual	−14.1	−5.7	35.4	−17.9	
	Voluntary	Count	56	94	4	1156	1311
		Expected count	330.3	187.7	386.6	405.1	1311
		Standardised residual	−15.1	−6.8	−19.5	37.3	

**Table 3 diagnostics-12-01806-t003:** True Positive, False Positive, True Negative and False Negative values ^1^.

	Voluntary	Assisted	Involuntary	Declined
True Positive	33	26	29	17
False Negative	4	6	4	12
False Positive	4	5	11	6
True Negative	90	94	87	96
Total	131	131	131	131

^1^ Participants contributed to a composite value for each clinical case narrative.

**Table 4 diagnostics-12-01806-t004:** Predictive validity calculations.

	Voluntary	Assisted	Involuntary	Declined
Predictive Accuracy	94%	92%	88%	86%
Sensitivity (True Positive Rate)	89%	82%	87%	59%
Specificity (True Negative Rate)	96%	95%	89%	94%
Positive Predictive Value (Precision Rate)	89%	84%	72%	74%
Negative Predictive Value	95%	94%	95%	89%
False Negative Rate (Miss Rate)	11%	18%	13%	41%
False Positive rate (Fall-out Rate)	4%	5%	11%	6%

**Table 5 diagnostics-12-01806-t005:** Pearson’s correlation coefficients for associations between the clinical features and case-specific accuracy.

	Case-Specific Accuracy%	Suicidality Features	Incongruence in Symptom Severity	Incongruence in Severity of Suicide Risk	Cognitive Impairment	Psychotic Features
Case-specific Accuracy%	1	−0.033	−0.170	−0.134	0.204	0.208
Suicidality features	−0.033	1	−0.502	0.440	0.002	v0.034
Incongruence in symptom severity	−0.170	−0.502	1	−0.257	−0.171	−0.143
Incongruence in severity of suicide risk	−0.134	0.440	−0.257	1	−0.184	−0.133
Cognitive impairment	0.204	0.002	−0.171	−0.184	1	0.026
Psychotic feature	0.208	−0.034	−0.143	−0.133	0.026	1

**Table 6 diagnostics-12-01806-t006:** Scores of the legal states and decision paths in the correspondence analysis ^1^.

Legal States	Mass	Score in Dimension 1		Score in Dimension 2	
		Legal state	Decision path	Legal state	Decision path
Assisted	0.252	0.408	0.409	1.265	1.271
Declined	0.144	1.679	1.685	−1.708	−1.717
Involuntary	0.295	0.215	0.216	0.414	0.416
Voluntary	0.310	−1.315	−1.320	−0.629	−0.632
Total	1.001				

^1^ Symmetric normalisation was applied.

**Table 7 diagnostics-12-01806-t007:** Krippendorff’s alpha values for the legal states and the decision paths.

Consistency Parameter	Participants	Alpha Coefficient	Bootstrapped Alpha Coefficient	95% Confidence Interval
Legal states	All participants	0.650	0.663	0.634–0.693
	General physicians-in-training	0.653	0.666	0.637–0.694
	Psychiatrists-in-training	0.644	0.855	0.754–0.952
Decision paths	All participants	0.426	0.447	0.424–0.471
	General physicians-in-training	0.430	0.451	0.426–0.477
	Psychiatrists-in-training	0.417	0.764	0.623–0.899

## Data Availability

The Research Ethics Committee that approved the research determines the limits on the availability of raw and processed data based on the merits of an application to gain access, the interests of stakeholders, and the mandates of research regulatory authorities. No special computer code or syntax is needed to reproduce analyses other than provided standardly in the SPSS-software program (version 27). Regarding the availability of research materials, the algorithm is available as a supplement to the article.

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
