# Peer review of "Algorithmic Assessments in Deciding on Voluntary, Assisted or Involuntary Psychiatric Treatment"

_diagnostics, 2022, doi:10.3390/diagnostics12081806_

Round 1
Reviewer 1 Report
The study reflected in the manuscript is well executed, designed and reported. The topic is highly relevant, especially in the field of Mental Health/Psychiatry. The research problem is properly formulated and the aim is stated clearly. The research designed is applicable and statistical methods are explained well. The conclusion is clear and the recommendations reflect the aim of the study.
However, there are some minor corrections to be done. These changes would increase the value of the study and the manuscript. The following minor corrections should be made:
1. Most of the references are between 1991 and 2015. Only three references are within the last 5 years. Please update with more recent references.
2. It is also recommended that a copy or a shortened version of the ILSAA be attached as an addendum to the manuscript, if possible.
3. Change "The" to "There" : P.1 paragraph 2 line 2.
Author Response
Comment 1. Most of the references are between 1991 and 2015. Only three references are within the last 5 years. Please update with more recent references.
Response: We have now added four recent publications (2022; 2021 and two of 2019).
2. It is also recommended that a copy or a shortened version of the ILSAA be attached as an addendum to the manuscript, if possible.
Response: The ILSAA is submitted as supplementary material to the article.
3. Change "The" to "There" : P.1 paragraph 2 line 2.
Response: This has now been corrected.
Reviewer 2 Report
An interesting article and completely revolutionary in its conception and execution.
Author Response
Comment: An interesting article and completely revolutionary in its conception and execution.
Response: We thank the reviewer for this comment.
Reviewer 3 Report
This is an article presenting an interesting topic for psychiatry. However, the way it is presented, it makes it very hard to understand the actual results. The researchers should try to present clearly their results not only in terms of statistical analyses but also in terms of clinical relevance and practical implications. As it is presented, it may be understood by somebody with a deep understanding of the relevant legal issues, but the journal to which it is submitted is not from South Africa.
I am unable to recommend publication unless the authors try to make the article easy to understand for somebody with average understanding of the topic, and who actually is interested in using this algorithm, especially as, as it is correctly stated in the article, it should be used not only by psychiatrists but also by physicians from other specialties.
Author Response
Comment: This is an article presenting an interesting topic for psychiatry. However, the way it is presented, it makes it very hard to understand the actual results. The researchers should try to present clearly their results not only in terms of statistical analyses but also in terms of clinical relevance and practical implications. As it is presented, it may be understood by somebody with a deep understanding of the relevant legal issues, but the journal to which it is submitted is not from South Africa. I am unable to recommend publication unless the authors try to make the article easy to understand for somebody with average understanding of the topic, and who actually is interested in using this algorithm, especially as, as it is correctly stated in the article, it should be used not only by psychiatrists but also by physicians from other specialties.
Response: We have now added to the introduction more context in which the study is situated as follows: "Whilst using diagnostic algorithms and prediction models is well-established in medicine [11, 12], no validated algorithm exists as far as we could establish for assessing whether psychiatric treatment should be provided on voluntary, assisted or involuntary legal basis".
Towards the end of the article we state "The algorithm is meant to assist and enable clinicians in making challenging, difficult, and non-exact assessments of the most suitable legal state for providing psychiatric treatments including hospitalisation. However, the contents of the algorithm were derived from requirements of MHCA in South Africa [13], and it may need adapting for other jurisdictions. To this end, the algorithm may serve as an exemplar".
We also state that this articles provides the scientific grounds for the use of the algorithm, as follows: "Supported by its validity and reliability, the algorithm may be used in practice at the clinical coalface where the assessments have proven to be challenging [3, 4, 7]. It may also be used in clinical training. Notwithstanding its utility in practice and clinical training, the results on the validity and reliability of the algorithm further warrant subsequent randomised controlled studies on whether it is more effective in practice and training than respectively standard assessments and training methods."
Thus, the results of our study (and its statistical emphasis) precede a focus on clinical practice. A proper focus on clinical practice would for example be a subsequent study reporting on the effectivity of the algorithm. We have now added to the Discussion section references (these are, 18 to 20) to similar validation studies that have a similar emphasis (on statistical validation) in providing the scientific basis for clinical assessment instruments (including the well-known SCID-5).
We are not sure whether the reviewer had access to the algorithm that was submitted as supplementary material as another reviewer seemingly did not. If this reviewer did not have access, it might have limited clarity on the clinical utility of the instrument.
Reviewer 4 Report
The following observations about the manuscript are made below:
- The abstract should be structured according to the sections of the manuscript, indicated the title of each section
- Keywords are recommended to be DeCS or MeSH
- Although the introduction is appropriate, it is recommended that data be provided on other similar studies that indicate what success rates have been obtained using these algorithms or what critical aspects they present, as well as their validation data, such as the ILSAA.
- In the research participant section, it is important to explain the total target population referred to and how they were accessed. That is, the total number of possible participants, divided by training and senior, and which access routes were carried out.
- For the validation of the clinical cases, was the concordance index found or was some type of Delphi-type technique used?
- Was the study approved by an ethics committee?
- Was a factor analysis of the ILSAA performed to check its validity?
- The results indicate: The 317 participants comprised of 300 senior medical students and 17 psychiatrists-in-training. But this is confusing. How many were students in training and how many were seniors?
- The gold standard should be clarified in order to assess its significance.
- In Table 1, the means must be accompanied by the standard deviations
- In table 3, indicate that the results correspond to the n of each item.
- In Table 6, the data in parentheses are standard deviations?
- In the discussion, the statements of the results obtained compared with more studies should be supported, whether these agree or disagree.
Author Response
See responses to comments in the attached document.

Round 2
Reviewer 3 Report
The authors have revised the manuscript according to the requirements of the reviewers. Is it is, it may be published.
Reviewer 4 Report
Following review of the authors' responses to comments on their manuscript, these are considered to have been included and taken into account. Only, it is recommended that in table 1, "standard deviation" be replaced by its acronym "sd" and with an asterisk indicate at the bottom of the table that it refers to that.